# The Driverless Bus: An Analysis of Public Perceptions and Acceptability

**M. Eugenia López-Lambas [1] and Andrea Alonso [2],*** 

[1]    Civil Engineering School, Department of Transport Engineering, Urban and Regional Planning, Universidad Politécnica de Madrid, 28040 Madrid, Spain; mariaeugenia.lopez@upm.es

[2]    School of Architecture, Department of Urban and Regional Planning, Universidad Politécnica de Madrid, 28040 Madrid, Spain

*    Correspondence: andrea.alonso@upm.es; Tel.: +34-9106-75096

**Abstract:** The development of autonomous vehicles (AVs) holds a high potential for improving security, reducing congestion, increasing fuel efficiency, and saving time. Various studies conducted on the implementation of AVs predict that fully autonomous vehicles will be available for the public in the 2020s. However, it will take another three decades, at least, for these vehicles and technologies to be accepted among the general masses and become reliable and affordable for use. Nonetheless, while a great deal has been stated regarding autonomous cars, little attention has been paid to autonomous public transport, more specifically, autonomous buses (ABs). The present report analyzed the psychological barriers preventing the complete implementation of ABs through data collected from focus group (FG) discussions. The main objective of the FGs was to determine the factors influencing perceptions regarding ABs and their acceptability. The most important factors from the positive side were the reduction of personnel costs, the potential to decrease congestion, waiting time at intersections, and reduced emissions. On the other hand, the most important negative factors were an increase in vehicle and infrastructure costs, safety risks under certain conditions (e.g., system failures, terrorist attacks, etc.), and the possible reduction of employment opportunities.

**Keywords:** autonomous vehicles; driverless buses; acceptability; focus group; barriers

## 1. Introduction

The automation of transport is expected to bring a shift in mobility. This will significantly affect the way of life for many people [1] and will serve as a key aspect determining the competitiveness of transport companies, affecting their future position in economic, environmental, and social spheres [2,3]. It is not surprising that the automation of transport has become a priority thematic area at the global level, which is reflected in several international research programs (e.g., Horizon 2020). A few of the benefits expected from the automation of transport are increased security and efficiency, major comfort for users and professionals, and enhanced social inclusion and accessibility [4].

In this context, project AUTOMOST (2016–2020) [5] was implemented. The project's aim is to analyze the technical and psychological barriers along the path towards implementation of autonomous vehicles, specifically, autonomous buses. The present report deals with the psychological barriers only, analyzed through data collected from focus groups (FGs).

Progress in the field of autonomous driving, which is understood as the capacity of a vehicle to drive by itself, is a fact. Niche manufacturers have estimated that by the 2020s, fully autonomous vehicles will be available on the market [6]. The development of this technology is fostered by European and other various international institutions due to the potential of this technology to improve security,

increase fuel consumption efficiency, and save time. Indeed, autonomous vehicles (AVs), as well as electrification and vehicle sharing, can be the next big change in the field of mobility [7].

However, while autonomous vehicles may be available on the market in the 2020s, it will take at least another three decades until these vehicles receive sufficient acceptance and become adequately reliable and affordable enough to account for 50% of the global fleet [4]. The barriers that are required to be overcome in order to introduce autonomous vehicles to the road and the consequences of this development are uncertain, and several researchers have analyzed this previously [2,4,8]. The positive and negative aspects of these reported technologies from these studies—and in certain others conducted on autonomous buses [9–11], a topic for which the literature is scarce [12]—are summarized in Table 1.

**Table 1.** Important aspects regarding autonomous vehicles.

| Autonomous Driving—General Aspects | |
| --- | --- |
| **Positive Aspects** | **Negative Aspects** |
| <ul><li>Savings on staff and driver costs.</li><li>Increased security and reduced risk of the most common accidents (lack of attention, safe distance, unsafe overtaking, etc.).</li><li>Reduced fuel consumption and less pollution generation due to the fact of more efficient driving.</li><li>Optimization of time, as there are no driving time losses (one may use that time to do other things).</li><li>Improvement in equity, as elders and children would gain accessibility, and it would be less limiting for people without a driving license.</li></ul> | <ul><li>Major investment required, as specific equipment, special vehicles, better infrastructure (connected roads), and maintenance would be required.</li><li>New risks would arise as there would be less safety and security in special situations, such as system failures, terrorist attacks, and situations requiring human-based decisions.</li><li>Automatization may reduce job opportunities.</li><li>Territorial aspects, for example, reductions in the transport costs might encourage people to reside further from their workplace, increasing urban sprawl.</li></ul> |
| **Autonomous Buses—Specific Aspects** | |
| **Positive Aspects** | **Negative Aspects** |
| <ul><li>Potential to increase reliability and punctuality (connected infrastructure, traffic signals prioritizing buses, etc.).</li><li>Potential to reduce staff costs (if a public service is involved, society may benefit).</li><li>Potential to increase investment (infrastructure and vehicles), and therefore, a possible multiplier effect.</li><li>Flexibility for changing routes, timetables, etc., as schedules do not depend on human drivers.</li></ul> | <ul><li>Non-payment of tickets would be difficult to control.</li><li>Lack of an individual to address in case of any incidence, such as buying a ticket, specific information requirements, specific passenger requirements, etc.</li><li>Possible reduction in passenger's safety/security perception as no individual would represent or assume the authority on board.</li><li>Driver's role as an information provider disappears.</li></ul> |

In addition to the studies mentioned above and summarized in Table 1, several other studies have been conducted regarding the actual and potential technical, administrative, or legal constraints and benefits of autonomous driving [13–15], or regarding the motivations for owning an autonomous vehicle [16]. However, limited focus has been paid regarding perceptions of autonomous vehicles, particularly regarding perceptions of autonomous buses, i.e., how individuals feel regarding the possibility of being transported by a driverless public transport means other than metro rail or taxi, or whether factors affecting the public perceptions of autonomous vehicles are similar to those affecting autonomous buses.

The fact is that the limited number of scientific studies conducted on autonomous buses deal only with the cost analysis of the services [12]. As stated by Shin et al. [15], limited data are available on how consumers value emerging vehicular technologies. According to the reviewed literature, no studies are

available which have used the FG technique in order to identify the factors influencing the adoption and usage of autonomous buses.

A focus on autonomous buses generates alternative perspectives. For instance, several researchers have predicted that autonomous cars will lead to a general reduction in the travelers' value of time, as their occupants will be able to perform other activities (leisure or productive) while traveling [17], in addition to the reduction of stress, among other advantages. However, this would not be the case with autonomous buses, as it could be argued that public transport is already offering these possibilities. Therefore, it all turns out to be about efficiency. It is in this context that the UITP's (Union Internationale des Transports Publics) position paper pleads for a future of shared mobility, with driverless car fleets reinforcing the public transport network, as illustrated in Figure 1. In this sense, autonomous public transportation can provide the key to a *sustainable mobility solution* [7].

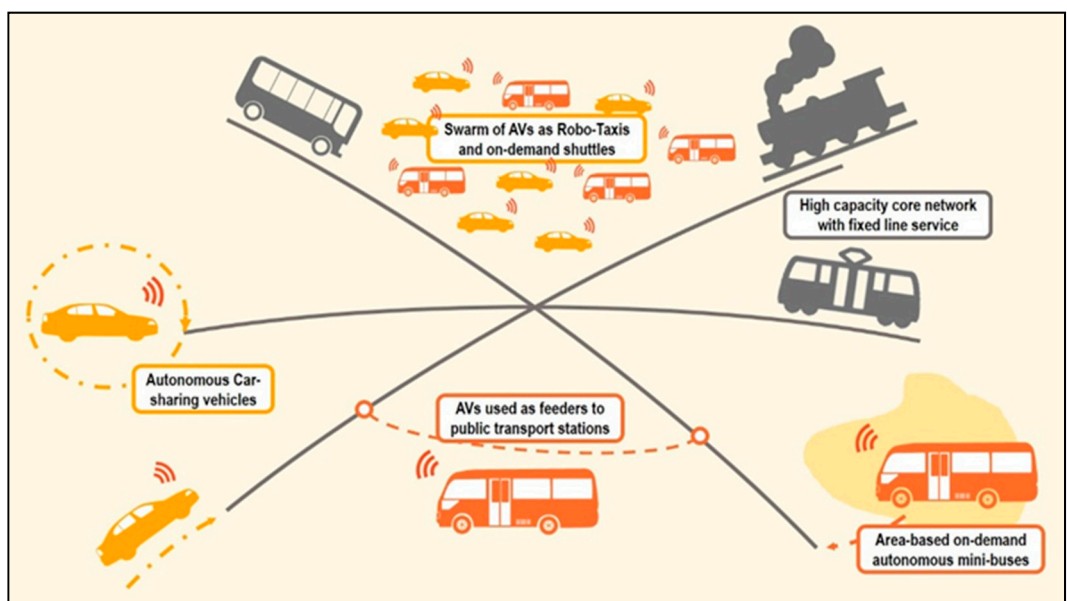

**Figure 1.** Possible applications of autonomous vehicles as part of a diversified public transport system; adapted Union Internationale des Transports Publics (UITP).

The present report describes the results of the two FGs which were developed in the framework of project AUTOMOST [5], with the objective of outlining all the aspects that could affect the implementation of autonomous vehicles, particularly autonomous buses, and exploring which of these aspects could be more interesting.

## 2. Methodology: The FG Approach

Focus groups may assist in capturing insights into several issues ranging from decision making to policy or service development [18], and, of course, behavior. In the AUTOMOST project [5], the researchers expect to determine the level of acceptability among individuals for an innovative technology applied to a means of transport other than a car, for example, buses. The objective of the project was to identify the main objections to the usage of autonomous buses and to generate ideas regarding methods to overcome these objections in order to design a strategy to introduce self-driving buses onto the market. The main difference between the FG approach and the one-to-one interview approach lies in the fact that while the latter aims to probe experience, an FG is intended to serve as a group of individuals who are able to generate ideas [19].

The FG approach may be defined as a semi-structured group session, moderated by a group leader, and held in an informal setting, with the purpose of collecting information on a designated topic [20]. Unlike one-to-one interviews, FG debates are more accommodating in that they allow for

the generation of novel ideas [19]. Focus group debates are used widely in different fields of research to obtain qualitative data [21].

Merton and Kendall [22] developed the FG technique as a key element for studying human behavior and sociology. Since the time it was originally developed, the FG technique has been used to conduct market analysis, evaluation of product acceptance, analysis of the structure and organization of companies, health research, analysis of social issues, etc. [23].

A focus group interview is a sort of group interview in which, owing to the communication among the participants, knowledge and data are generated [24]. Interaction among the participants is the key component of this technique and another main factor which differentiates FGs from one-to-one interviews. This type of group exercise is quite effective in assisting individuals to explore and clarify their thoughts and opinions.

Nevertheless, the analysis and utilization of FGs is a complex task, and although it generates qualitative information which is relevant and innovative, it does not offer statistically significant results [24], which is the reason that the application of this technique is not always advisable. However, as stated by Krueger and Casey [18], FGs are ideal when exploring perceptions, attitudes, and thoughts among individuals regarding a plan, policy, or tendency which will affect them in the future, and also when searching for novel ideas, concepts, and issues related to a new topic. Both of these objectives were fulfilled and are included in the present report, i.e., a better understanding of the perceptions among people regarding the implementation of autonomous cars and buses in a manner that raises novel questions and factors related to this relatively new topic.

The research described here was based on two FGs, the same number of FGs used by other authors to support their own qualitative analyses [25–27]. In order to correctly develop both of the FGs, they were organized in accordance with the following guidelines provided by Kitzinger [24], and Krueger and Casey [18]:

- It is advisable to organize the FG with the number of participants ranging between 6 and 10 [20]. This number should represent the target population to the fullest extent possible. Besides, it is convenient that the focus group be diverse in terms of the variables that affect the topic under analysis the most. In the case described in the present report, each group consisted of eight participants, and the variables considered to ensure diversity were age and gender; according to different authors [28], these are the characteristics that influence the acceptability of autonomous driving the most. Each group consisted of a man and a woman aged between 20 and 30 years, between 30 and 40 years, 40 and 60 years, and 60 and 80 years.

- The sessions should be conducted in a relaxed atmosphere, where the participants feel at ease, with all of them seated in a circle for convenient interaction. One or two coordinators or moderators should introduce the topic, encourage communication among the participants, lead the discussion, and raise questions. Two moderators intervened in the FGs described in the present report. A brief introduction of the subject was presented to the group at the beginning of the session, in order to provide information on the current situation, while explaining that they could ask for clarifications.

- After the introduction of the subject, all the participants should talk freely regarding what they consider appropriate, to encourage the emergence of novel concepts and to determine the priority issues required to be taken into consideration. Nevertheless, in order to gain the best possible results from the FG, the moderator(s) should raise those important issues that have not been brought up naturally during participant discussions. Besides, the moderator(s) must attempt to have everyone engaged, preventing a particularly dominant participant from overtaking the conversation and crowding out other participants [19]. In the case included in the present report, the moderators used notes based on the information provided in Table 1. Therefore, the factors already analyzed by existing literature on autonomous driving (described in Section 1 and summarized in Table 1) were the starting points for the FG. This should ensure the added value of the results compared to other studies.

- Ideally, the sessions should be recorded and subsequently transcribed. In the case at hand, the sessions were recorded, while a person took notes that were later transcribed. The sessions lasted for 1.5–2 h. Once the files with the recordings were extracted, they were transcribed word by word in order to better grasp the emotions and the intensity of the comments, and the different ways of thinking, which would otherwise be impossible to grasp.
- Once the FG plans have been conducted, consideration of whether the saturation point has been reached should be done [18]. According to the authors in Reference [18], saturation is a term used to describe the point where a range of ideas have been raised on a recurring basis and no new information appears. If this point has not been reached, it would be necessary to program more FGs. However, it is always advisable to carry out more than one FG in order to conduct an analysis across groups. In the case described here, two FGs were planned in principle, the first one in Madrid and the second one in Malaga. Most ideas and thoughts were raised in Madrid and corroborated in Malaga. Actually, in this second FG, several new concepts appeared but it mostly consisted of repeating the same notions in different words. Therefore, it was concluded that the saturation point was reached in Malaga and there was no need to program a new FG.

## 3. Results

All the concepts that emerged on the basis of the transcriptions were defined and conceptualized. Automated driving is a novel and futuristic topic. The discussions focused on the effects (positive or negative) that autonomous driving technology would cause and the barriers for the development of this technology. Tables 2 and 3 list the positive and negative effects of autonomous driving, divided into five categories: environmental, psychosocial, socioeconomic, effects related to the vehicle or the infrastructure, and the effects related to novel business models associated with autonomous vehicles. Table 4 describes the relevance of each effect on autonomous vehicles. Table 5 lists the barriers along the path to the development of autonomous vehicles in general and autonomous buses in particular.

**Table 2.** Potential positive effects of autonomous driving from the focus groups (FGs).

| Positive Effects (+) | | Concept |
|---|---|---|
| Environmental (E) | Energy saving | Autonomous driving would be energy-wise more efficient and controlled. Vehicles would be programmed to minimize energy consumption, employing eco-driving patterns. |
| | Clean energy resources | When renewing the vehicle fleet, these would operate using electricity or cleaner fuels. |
| | Minimization of damages to wildlife/flora | Vehicles would drive in an environmentally friendly manner, and since the vehicles would be equipped with sensors, these would probably reduce collisions with animals. |
| Psychosocial (PS) | Comfort | It would eliminate driving activities for people. |
| | Reduced stress | It would reduce the stress generated in certain people as a result of driving activities, and also the general stress of driving in traffic jams. |
| | Accessibility | It would allow the extension of usage of vehicles to people who cannot drive, such as the elderly, children, or the disabled. |
| | Avoidance of human errors | Human errors, such as lapses of concentration or vision, or perception difficulties would be avoided. |
| | Regulatory compliance | The traffic regulations that are currently transgressed by human drivers would be met; the signals and speed limits would be respected and there would be no driving under the influence of alcohol. |
| | Reduction in traffic-based accidents | The autonomous vehicles would be able to better detect the presence of people and obstacles existing on the road owing to the installed sensors (without human errors) and would be programmed to act rapidly, avoiding certain accidents. This factor is related to the previous two factors. |

**Table 2.** *Cont.*

| Positive Effects (+) | | Concept |
|---|---|---|
| Socio-economic (SE) | Driving without physiological limitations | Autonomous vehicles would not require sleep or rest and lack all the physiological needs of a human driver, which would lead to efficiency and effectiveness in driving. More flexibility for bus companies to plan schedules. |
| | Better utilization of time | It would increase the usefulness of time by saving driving activities. The occupants of the vehicle would be able to work or rest while traveling. |
| | Skilled employment | The demand for skilled workers for tasks such as sensor design and driving parameters and procedures would increase. |
| Infrastructure and vehicle ((IV) | Maintenance improvement | The vehicle would be responsible for detecting the deficiencies in maintenance; it could be equipped with devices that would detect failures or the requirements of the vehicle. |
| | Better road sign management | Owing to V2V and V2I communication, the signals, roundabouts, and other elements of the infrastructure could be dispensed with. |
| Novel business models (NBMs) | Enhanced collaborative economy | The vehicle should not be parked while waiting for a passenger to use it, and it should only be available at the time when one requires it. It could go and seek passengers by moving toward them. The time when nobody is using it, the vehicle could perform other services for the other passengers instead of standing idle. |
| | Less drivers | It would save the costs of hiring drivers. |
| | Less owned cars | By promoting collaborative economy, the requirement to own a car would be reduced, thereby resulting in fewer vehicles on road. A more equitable system would be generated with the better efficient use of vehicles and infrastructure, utilizing the advantages and the capabilities of both. |
| | Reduced city parking | The vehicles would be spending more time moving than being parked. In addition, it would no longer be necessary to park the car close to the origin or destination of the trip, since the car could pick up the passenger. Therefore, there would be fewer cars halted, and that will be in parking lots (not necessarily located in the urban center or close to the passengers). |
| | More efficient freight transport | In regard to freight, autonomous vehicles have high development potential and advantages that could affect the management of all transport; advantages such as night driving or driving at the times when there is no traffic would free the roads at peak hours. In addition, certain advantages associated with economies of scale could be acquired, for example, platooning. |

**Table 3.** Potential negative effects of autonomous driving (FGs).

| Negative Effects (−) | | Concept |
|---|---|---|
| Environmental (E) | High energy consumption by sensors | Both vehicles and infrastructure would require being equipped with multiple sensors and devices for detection and perception, which would consume a large amount of energy. |
| | Land consumption | If automation encourages the use of private transport and reduces the use of public transport, the problems of space occupation per passenger would not be solved and could even worsen. |
| Psychosocial (PS) | Serious potential failures | Although human error is more frequent, it is possible for the people to perceive it and rectify the error. However, when the failure is computational, it is more difficult to be rectified. |
| | Impossibility of improvisation | In the case of unexpected, unusual, or unforeseen events, vehicle programming could collapse without seeking alternatives to resolve the situation (for example, cut roads or saturated intersections) |
| | Accidents with greater impact | There would probably be fewer accidents, although these would be more impactful. Psychologically, it is convenient to assume human error rather than the error of a machine. One is inclined to accept an accident better when there is a physical person to blame. |

**Table 3.** *Cont.*

| Negative Effects (−) | | Concept |
|---|---|---|
| Socioeconomic | Potential economic inequality | Not everyone could own an autonomous car, as it would be a luxury item that only the people with high purchasing power would have access to. This would increase the gap among classes in society. |
| | Fewer driver jobs | If driving is automated, a number of jobs for bus, taxi, or truck drivers would be reduced. In addition, there would be a reduction in the employment opportunities for all personnel associated with driving schools and learning to drive in general. |
| | Congestion problems | If automation encourages the use of private transport and reduces the use of public transport, it would increase traffic congestion. This would have negative outcomes in terms of consumption of time and resources. |
| Infrastructure and vehicle (IV) | Major investments required in infrastructure | Infrastructure would have to be conditioned and connected for the feasibility of safe autonomous driving (via V2I communication). This would require large investments in quality infrastructure. |
| | Inadequate road infrastructure in rural and/or remote areas | It would not be feasible to provide connected and quality infrastructure to the entire territory. Probably, rural, remote, or inaccessible areas would not be reached. |
| | Potential meteorological failures | Adverse meteorological or other conditions could threaten the reliability of driving. The role of sensors in autonomous driving is fundamental. |
| | Demanding maintenance of sensors | It would be necessary to perform frequent and constant maintenance of the sensors in order to ensure that they work properly. |
| Novel business models (NBMs) | Competition with public transport | The collaborative economy model that is expected to promote autonomous vehicles would result in service with characteristics similar to those of public transport, such as being affordable and accessible. Therefore, these models could compete with collective models, which would have negative outcomes for the economic sustainability of public transport. This would also be related to other problems such as space occupation and congestion. Cybersecurity of an autonomous car is a relevant issue. Computer attacks affect all levels and all big companies. Therefore, it is possible that they could also affect autonomous driving. |
| | Risk of hacking | The danger, in this case, would be that the physical control of the car and its elements (i.e., doors, windows, etc.) would be in the hands of third-party people. Another related risk would be the loss of privacy owing to the high amount of data that the system would accumulate. |

**Table 4.** The relevance of the effects of autonomous buses.

| Relevance of the Effects of Autonomous Buses | | |
|---|---|---|
| Very relevant | Energy saving (ES+) | As in the case of cars, this would also be a more efficient way of driving. In addition, if the implementation of autonomous buses manages to promote collective transport, the global energy savings would be greater. |
| | Serious potential failures (PS−) Accidents with greater repercussion (PS−) | Any accident that occurs on an autonomous bus would have greater repercussions, as it would affect a greater number of people. |
| | Drivers with no physiological requirements (SE+) | From the perspective of a bus operator, drivers are assets; if they do not require rest or sleep, the efficiency is higher. |
| | Encouraging skilled employment (SE+) | Bus operators and the related companies would require qualified personnel for monitoring and control functions. |
| | Fewer driver jobs (SE−) | The impact would be on the bus operators. |
| | Better road sign management (IV+) | Traffic management associated with automation is based on the communication among different devices (I2V). This would allow the prioritization of the circulation of buses, encouraging their use. |
| | Potential meteorological failures (IV−) | The fact that autonomous buses constitute a collective mode of transport renders this effect more relevant when dealing with buses. |
| | Fewer drivers (NBM+) | This effect is highly relevant for the bus operators as it alters their cost structure. |
| | Risk of hacking (NBM−) | The failures in security would have more repercussions in case of buses (as more people would be involved). Besides, bus operators are more likely to be hacked than individual car owners |

**Table 4.** *Cont.*

| Relevance of the Effects of Autonomous Buses | | |
|---|---|---|
| Relevant | Clean energy sources (E+)<br>High energy consumption by sensors (E−)<br>Avoidance of human errors (PS+)<br>Regulatory compliance (PS+)<br>Reduction in traffic-related accidents (PS+)<br>Impossibility of improvisation (PS−)<br>Maintenance improvement (V+)<br>Major investments required in infrastructure (V−)<br>Inadequate road infrastructure in rural and/or remote areas (V−)<br>Demanding maintenance of sensors (V−) | General effects are extensible to driving automation. |
| Little relevant | Minimization of damages to wildlife/flora (E+)<br><br>Fewer owned cars (NBM+)<br><br>Comfort (PS+)<br>Reduced stress (PS+)<br>Accessibility (PS+)<br>Better utilization of time (SE+)<br>Economic inequality (SE−) | Due to their driving styles and the places which they travel to, buses produce less damage to the flora and fauna.<br>The effect does not refer to buses, but to other business models, such as shared vehicles. However, since autonomous buses may encourage the use of public transport, it would reduce the number of vehicles.<br>In the case of buses, passengers do not have to drive and, therefore, there are no evident changes in comfort or stress due to the savings in driving activities.<br>Buses are already accessible to those who cannot drive.<br>Passengers of a bus do not drive; autonomous driving does not increase their time value.<br>This effect is relevant for car ownership and not for public transport. |
| Irrelevant | Congestion problems (SE−)<br>Enhanced collaborative economy (NBM+)<br>Reduced number of city parking lots (NBM+)<br>More efficient freight transport (NBM+)<br>Competition with public transport (NEG−−) | Collective transport does not contribute to congestion.<br><br>These effects do not refer to buses, but to other business models. |

**Table 5.** Potential barriers to the development of autonomous driving (FG).

| Barriers | | Concept |
|---|---|---|
| Perception | Sense of not having control | The feeling of not being able to control the driving personally may be negative for security. In the case of buses, passengers never control the vehicle, although they perceive insecurity only if there is no person handling the controls. |
| | Lack of experimentation | In the beginning, experimentation would be lacking for autonomous driving. Nevertheless, this effect is not perceived as crucial in the case of buses, as they will always follow the same route, which would be pre-determined, making the perception of security greater. |
| | Road automation is more complex than the other means of transport | Transport means including trains or planes are automated. Even then, they are not perceived as unsafe, unlike road transport which is perceived as a much more complex system to automate due to the interference of other vehicles and road users. |
| Legal | Difficult to assign responsibility in case of an accident | In the event of an accident, it would be more complex to assign and divide responsibilities among the manufacturer, the owner (in case of poor or inadequate maintenance), and other road users (inadequate behavior), etc. |
| | Complex management of insurance | Owing to the complexity of assigning responsibilities in the event of an accident, the insurance schemes and risk-sharing would become complicated tasks. |

**Table 5.** *Cont.*

| Barriers | | Concept |
|---|---|---|
| Transition | Difficult to switch from automatic mode to manual mode | If the vehicle goes in the automatic mode, and in an emergency situation, asks the driver to assume control, it is possible that the driver is not trained for it or is distracted, resting, asleep, or even drunk at that particular time. |
| | Complex management of a mixed fleet (autonomous and conventional vehicles) | When the whole fleet is automatic, there would be fewer problems. However, the coexistence of manual and autonomous vehicles would be complicated. |
| | Autonomous cars are quite expensive | Initially, until the technology becomes popular, autonomous vehicles would be quite expensive and not quite affordable. |
| Mobility | Urban areas are complex | In comparison to other areas (motorways, high-quality roads, etc.), the urban environment would be more complex, owing to the number and diversity of road users and signals. |
| | Rural and remote areas are complex | In comparison to the other areas, the rural environment would be more complex, due to the difficulty encountered in adapting the infrastructure to the conditions required for autonomous driving and V2I communication. Investment in rural, peripheral, or less busy areas are always lower, and it is difficult for maintaining the right conditions for autonomous driving in rural infrastructure. |

| Specific Barriers to Autonomous Buses | | Concept |
|---|---|---|
| Absence of a human driver | On-board staff to resolve or report incidents | It would be necessary to have staff with technical knowledge sufficient to encounter problems that may arise during a trip or to report incidents, in order to avoid service interruptions as a result of unforeseen events. |
| | Lack of staff to provide information | The driver knows the route, stops, and the schedule. If in doubt, the passengers may ask the driver for information. This role of drivers would be lost with autonomous buses. |
| | Lack of a representative of the competent authority | The driver is the representative of the authority on board. Therefore, the absence of a driver could result in an increase in the probability of criminal actions. |
| | Higher risk of non-payment or non-compliance | With nobody to control the passengers boarding the bus, there would be an increased risk of non-payment and non-compliance with the rules. |

Figure 2 presents the importance given to each of the effects associated with autonomous driving according to the number of times the effect was mentioned during the sessions. It is noteworthy that much importance was given to the psychosocial effects associated with the compliance with rules, which indicated that a greater reliance was placed in machines rather than in people at the time of execution and application of standards (e.g., obeying speed limits and traffic lights). It is also believed that there would be fewer accidents because of a reduction in human errors. Among the disadvantages, the inability to improvise appears to be the most significant. It is also worth noting that the importance was given to novel business models, especially to the fact that autonomous driving would encourage collaborative economy, which is a concept that is becoming increasingly important nowadays. Environmental effects attracted few comments in general, although it is believed that the implementation of autonomous vehicles would have positive outcomes in general. On the one hand, it would promote car fleet renewal, favoring the use of cleaner energy resources. On the other hand, it would lead to more efficient driving, reducing the energy consumption per kilometer.

Figure 3 depicts that the socioeconomic aspects—especially the probable reduction in the number of jobs for drivers—which were among the most important negative attributes stated by the participants in the FGs. Nevertheless, it was agreed in general that these losses in jobs could be conveniently compensated through the creation of skilled employment (Figure 2).

The importance of each barrier along the path of the development of autonomous driving are listed in Figure 4. The fear of a new technology that has barely been proven appeared to be the most important barrier. This barrier was well aligned with the other recurrent barriers, such as the feeling of not having control over the driving. Other relevant barriers were those associated with the transition between manual and autonomous vehicles. It was believed that once the entire fleet is autonomous, driving will be safer. However, the coexistence period—where both autonomous and manual vehicles would be circulating together on the road—generated uncertainty, and was, therefore, associated with unsafety and compatibility issues.

Certain legal barriers were stated during the FGs, such as those associated with responsibilities in the event of an accident or those associated with insurance complexities. However, these barriers were not considered crucial, as it was believed that these would be resolved as the technologies evolved. Even the barriers associated with mobility did not appear to be noteworthy. As far as autonomous buses were concerned, the barriers were related to the absence of someone who could physically impose authority, in enforcing payment, informing, or preventing security attacks.

In order to obtain the results that were reported, the FGs held in Madrid and Malaga were jointly addressed. After processing the interventions of each of the participants, 413 relevant comments that fulfilled the objective of the present report were obtained (Figure 5). Most of these comments (64%) referred to the effects that the autonomous vehicles would have in the future, the positive ones being more frequent (37%) than the negative ones (17%). Forty-six percent of the comments were focused on the barriers along the path of development of autonomous vehicles, especially the barriers related to the perceptions of the lack of safety.

Figure 5 also reveals that the psychosocial and socioeconomic effects of autonomous driving were the most recurrent ones. In addition, autonomous vehicles were positively associated with novel business models and a collaborative economy. Appendix A provides details regarding the personal characteristics of the individuals who made reference to each of the concepts with their comments.

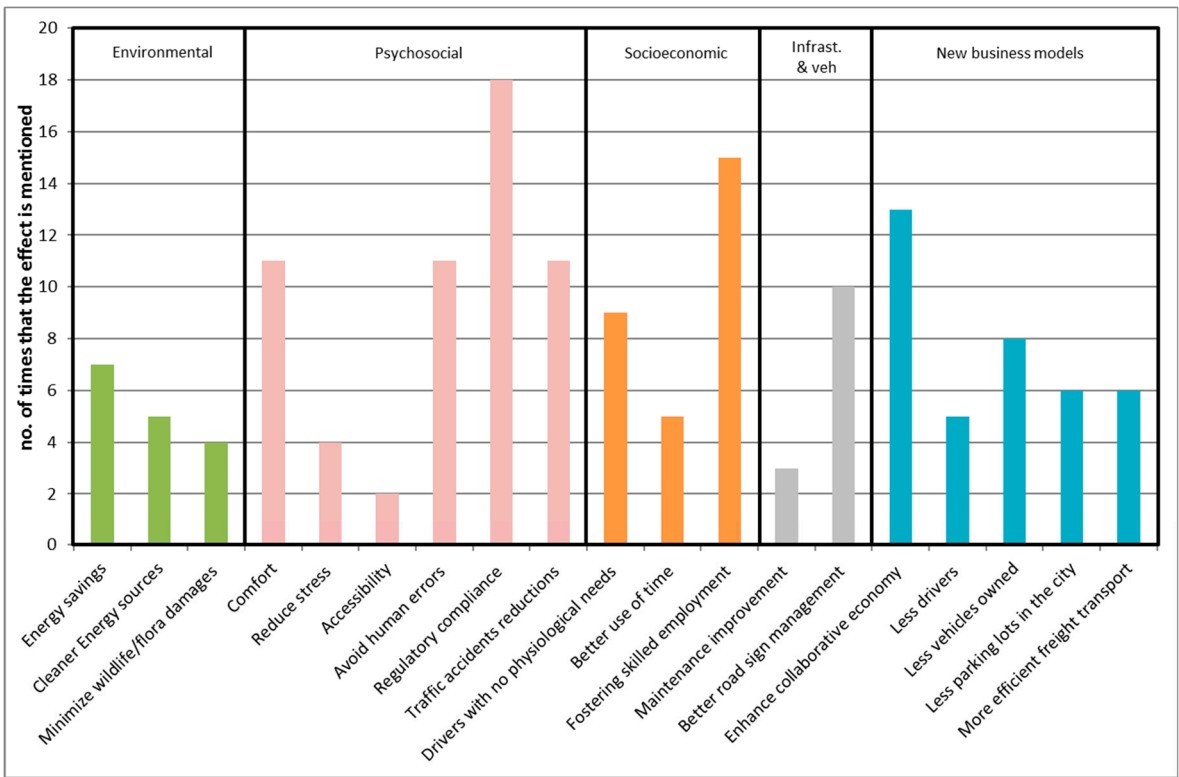

**Figure 2.** The number of times each positive effect was mentioned in the FGs.

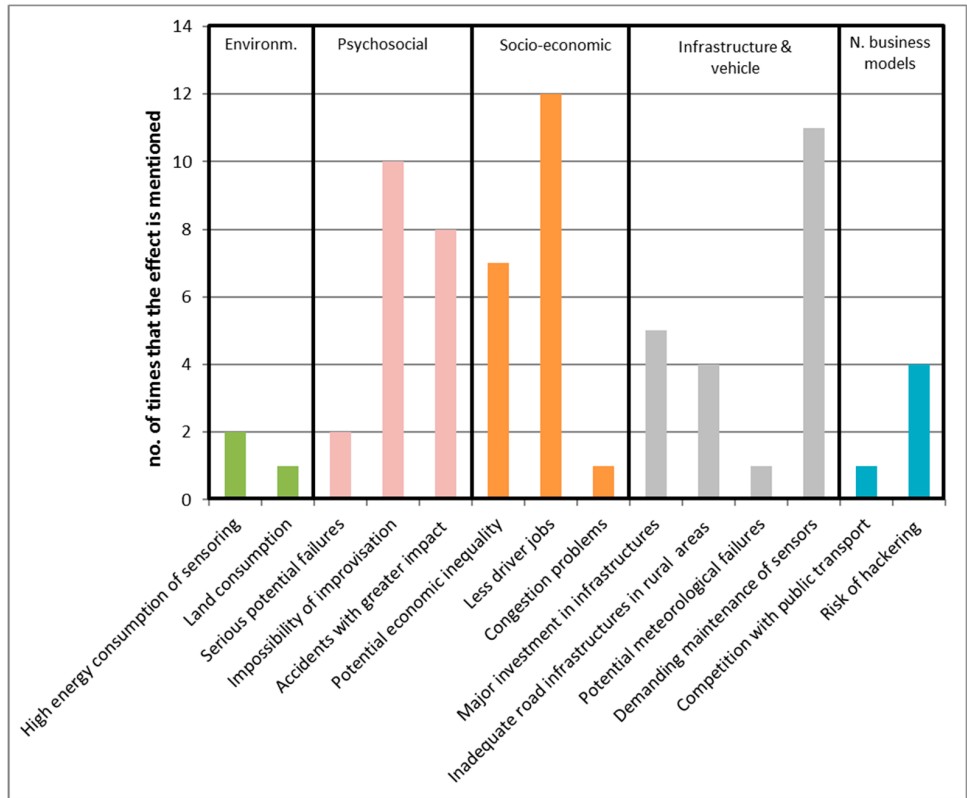

**Figure 3.** A number of times each positive effect was mentioned in the FGs.

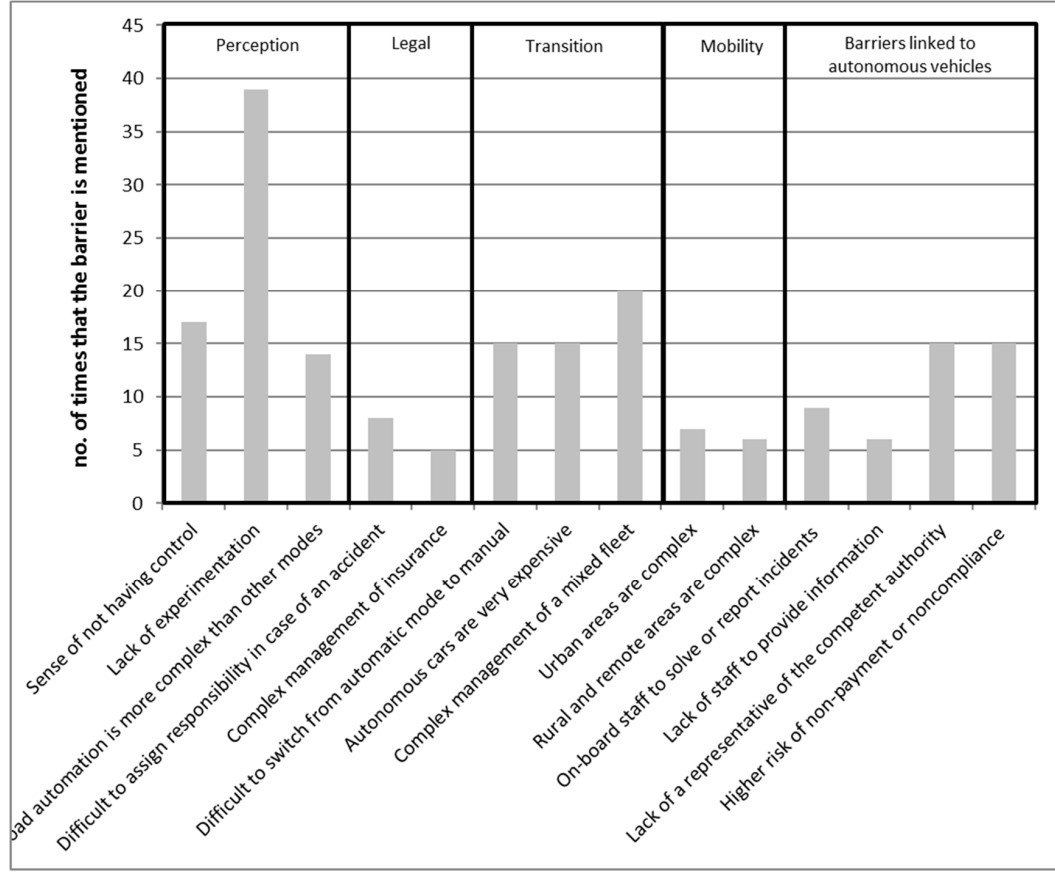

**Figure 4.** The number of times each barrier was mentioned in the FGs.

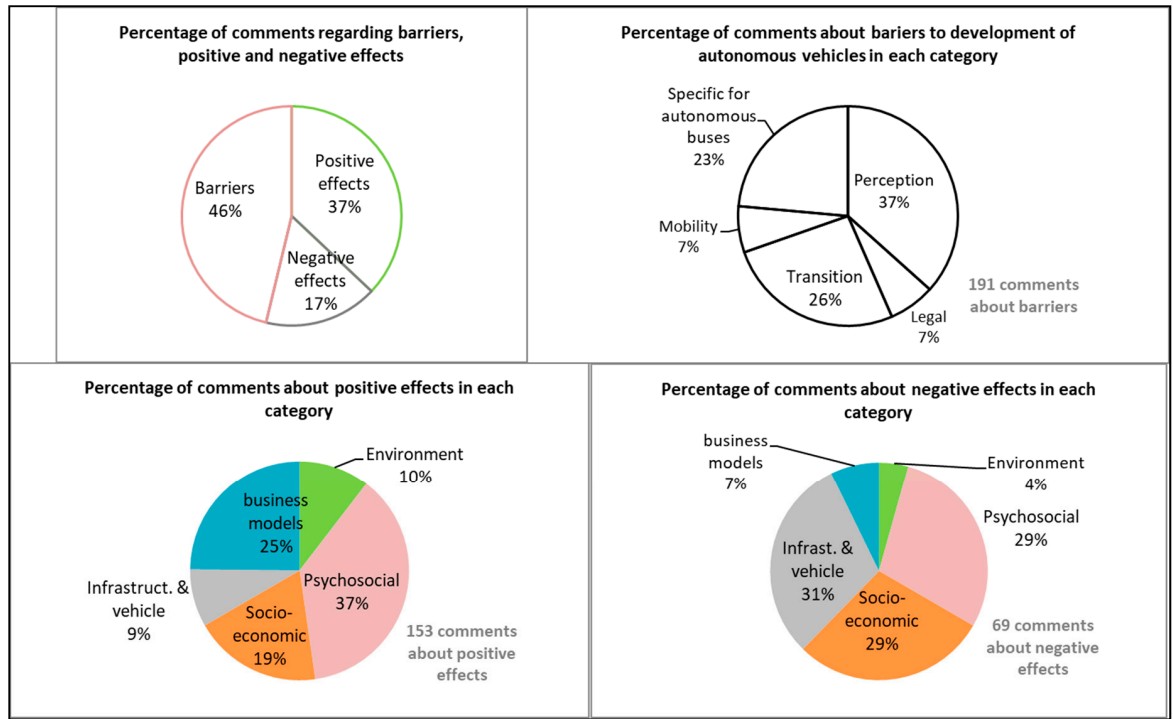

**Figure 5.** Percentage of comments that referred to the different concepts during the development of the FGs.

## 4. Discussion of Results

The added value of the results presented in comparison to the existing literature (summarized in Table 1) relies on results including the definition of all the concepts surrounding the implementation of AVs and ABs and their classification, in addition to the definition of novel concepts (highlighted in grey in Tables 2 and 3). Although the research was developed in the framework of a national project, the outcomes obtained are universally applicable and help to understand the barriers to the development of autonomous driving. Furthermore, autonomous buses constitute a topic that has been barely addressed in the literature [12]. The few existing studies are generally focused on psychological barriers [9,10], while this paper offers a more holistic approach.

In any case, there is a need to reflect on the possible limitations and potential extensions of the study presented. In the context of this research, the realization of more FGs would probably not have added new findings, as the saturation point was reached in the second FG. However, the FGs were developed in two Spanish cities and it is possible that if the experiment was repeated in other countries, participants would insist differently on each effect and barrier. For example, in a country such as the USA, which has more permissive laws regarding AVs and which has more experience in testing these technologies [29], perceptions and psychosocial barriers would probably not seem so important (Figure 5). Finally, it should be noted that most effects and barriers arise due to the mistrust that new technologies produce. Therefore, it would be convenient and interesting to repeat the study in a few years and analyze the outcomes again, when AVs have been more tested and developed.

## 5. Conclusions

In the present report, the results of two focus groups conducted in Malaga and Madrid were presented in order to define and evaluate the barriers and potential consequences of autonomous driving. The methodology was proven as effective in determining the key factors surrounding the introduction of autonomous vehicles. The added value of the present report compared to the existing literature is that this report conceptualized all of these factors and their classification into different categories. A convenient structure of the effects and barriers was generated, which assists in understanding the

possible consequences of autonomous driving in all areas. In addition, the report illustrates the most positive aspects (e.g., regulatory compliance or promotion of collaborative economy) as well as the most important barriers in the path of expansion of this technology (e.g., lack of experimentation or the complex management of a mixed fleet). However, the good news here is that these constraints can be overcome as the technology begins to penetrate the market [30].

Finally, in the present report, a topic barely addressed in the existing literature was analyzed, i.e., autonomous buses and their acceptability. All the concepts analyzed will be important for the implementation of autonomous cars and buses and must be considered in the marketing or promotional strategies for these technologies. Given that individuals are still less willing to ride in driverless vehicles than those with human drivers [10], the outcomes of this paper are clearly relevant.

The present report opens the scope for novel research works in this area of study. With the identification and conceptualization of the major concerns and interests perceived by people and the structuring of their perceptions performed in a logical manner, it would be interesting to conduct massive surveys among the general public or interviews with experts in the field. This would assist in prioritizing and evaluating, in a statistically significant manner, the most important barriers or the most worrying effects along the path of the development of autonomous driving.

**Author Contributions:** M.E.L.-L. organized and coordinated the FGs; A.A. collaborated in the moderation of FGs; A.A. analyzed the data and extracted the main concepts regarding autonomous vehicles and buses; M.E.L.-L. and A.A. wrote the manuscript and extracted the main conclusions.

**Funding:** This research was funded in the framework of the project AUTOMOST- GUIADO AUTOMATIZADO PARA SISTEMA DE TRANSPORTE DUAL, EXP 00093152/TIC-20160172.

**Acknowledgments:** The authors would like to thank Pablo Cidon for his work with the transcriptions of the FGs. We also acknowledge the collaboration of Grupo AVANZA to develop the FGs, and especially Virginia Patón, José Moreno, and Rafael Cortés.

**Conflicts of Interest:** The authors declare no conflict of interest

## Appendix A

The table in Appendix A shows the percentage of comments about each concept made by women, younger than 40 years, and in the FG held in Madrid.

**Table A1.** Percentage of comments about each concept made by each profile.

| | | Barriers and Effects | % Women | % <40 Years | % Madrid |
|---|---|---|---|---|---|
| Positive effects | E | Energy saving | 29% | 29% | 71% |
| | | Clean energy sources | 20% | 20% | 40% |
| | | Minimize wildlife/flora damage | 75% | 75% | 100% |
| | PS | Comfort | 45% | 27% | 36% |
| | | Reduce stress | 75% | 0% | 75% |
| | | Accessibility | 100% | 0% | 50% |
| | | Avoid human errors | 27% | 27% | 64% |
| | | Regulatory compliance | 28% | 33% | 83% |
| | | Reduction in traffic accidents | 27% | 45% | 64% |
| | SE | Drivers with no physiological needs | 56% | 67% | 100% |
| | | Better use of time | 60% | 20% | 100% |
| | | Fostering skilled employment | 40% | 53% | 60% |
| | IV | Maintenance improvement | 67% | 100% | 100% |
| | | Better road sign management | 20% | 80% | 90% |
| | NBM | Enhance collaborative economy | 38% | 54% | 69% |
| | | Less drivers | 20% | 80% | 80% |
| | | Less owned cars | 38% | 38% | 50% |
| | | Reduce city parking lots | 50% | 50% | 33% |
| | | More efficient freight transport | 0% | 33% | 100% |

**Table A1.** *Cont.*

| | | Barriers and Effects | % Women | % <40 Years | % Madrid |
|---|---|---|---|---|---|
| Negative effects | E | High energy consumption by sensors | 0% | 0% | 100% |
| | | Land consumption | 0% | 100% | 100% |
| | PSIPS | Serious potential failures | 50% | 50% | 100% |
| | | Impossibility of improvisation | 100% | 70% | 50% |
| | | Accidents with greater impact | 75% | 38% | 63% |
| | SE | Potential economic inequality | 57% | 57% | 57% |
| | | Fewer driver jobs | 58% | 58% | 50% |
| | | Congestion problems | 0% | 100% | 100% |
| | VSE | Major investment in infrastructures | 20% | 60% | 0% |
| | | Inadequate road infrastructures in rural and/or remote areas | 75% | 100% | 75% |
| | | Potential meteorological failures | 100% | 0% | 0% |
| | | Demanding maintenance of sensors | 27% | 55% | 91% |
| | IV | Competition with public transport | 100% | 100% | 100% |
| | | Risk of hacking | 25% | 25% | 0% |
| Barriers | Perc. | Sense of not having control | 47% | 18% | 88% |
| | | Lack of experimentation | 51% | 31% | 72% |
| | | Road automation is more complex than other means of transport | 36% | 36% | 71% |
| | Leg. | Difficult to assign responsibility in case of an accident | 38% | 13% | 88% |
| | | Complex management of insurance | 60% | 20% | 80% |
| | Tran. | Difficult to switch from automatic mode to manual | 40% | 47% | 93% |
| | | Complex management of a mixed fleet (autonomous and conventional vehicles) | 45% | 65% | 80% |
| | | Autonomous cars are very expensive | 33% | 67% | 53% |
| | Mob. | Urban areas are complex | 71% | 14% | 86% |
| | | Rural and remote areas are complex | 83% | 100% | 33% |
| | Buses | Personnel on board necessary to solve/report incidents | 56% | 67% | 67% |
| | | Lack of staff to provide information | 33% | 83% | 17% |
| | | Lack of a representative of the competent authority | 27% | 33% | 67% |
| | | Higher risk of non-payment or noncompliance | 53% | 73% | 33% |

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
