# Peer review of "The Driverless Bus: An Analysis of Public Perceptions and Acceptability"

_sustainability, doi:10.3390/su11184986_

Round 1

Reviewer 1 Report

In this report, the authors defined and evaluated the barriers preventing the complete implementation of autonomous buses and their acceptability through the development of focus groups. this report conceptualizes all of these factors and their classification into different categories, which is an advantage compared to other literatures. The positive and negative factors describbed in this report should be useful when considering developing marketing and promotional strategies for AB.

Overall, this report is well organized, the reviewer has comments:
1. Line 28: ‘bus’ is not a good word for keyword.
2. ‘Autonomous driving’ or ‘automated driving’, please be consistent.
3. Line 162: ‘… that autonomous driving technology would cause and the barriers in the path of development of this technology’. Please double check this sentence. It is hard to understand.
4. Line 273: ‘However, playing knowledge a crucial role playing knowledge a crucial role in individual’s choice decision making, ….’. Please double check this sentence.
5. Question about this report. The results were obtained based on two FG developed in two places. Is it possible that if one more FG is going to be conducted in another place or in one or several years later, some of the results will be changed significantly? If yes, how to monitor this changes?

Author Response

The authors would like to thank the reviewer its valuable comments. All of them have been considered and the paper has changed accordingly. All changes and added text have been highlighted in red. Below you can find the responses to each comment

The key words have been changed, bus is not a keyword by itself anymore. Now the word is driverless buses (line 28). We hope that the new list of key words is suitable. We have homogenized the report. Now it is more consistent (we have changed automated by autonomous in all cases: lines 42, 43, 133, 206, 213 and 289; Tables 3, 4 and 5) The phrase which in the previous version of the paper was in line 162 has been rephrased again (now line 169-171). We hope it is easier to understand now. The phrase which in the previous version of the paper was in line 273 has been written again (now line 169-171). We hope it is easier to understand now. The fifth comment is especially interesting. We have added some new references and better justified the number of FG developed in this research (lines 123 and 124). In this regard, we have added also some reflections for future research, future works or to expand the study presented. These reflections can be found in a new section: Section 4 (Discussion of results), in lines 263-281.

Reviewer 2 Report

sustainability576969

This paper opens up the scope for novel research works in this area of autonomous vehicles.

The present report analyzed the psychological barriers preventing the complete implementation of autonomous vehicles through the development of focus groups

The added value of the present report compared to the existing literature is that this report conceptualizes all of these factors and their classification into different categories.

The results of two focus groups developed in Malaga and Madrid have been presented, in order to define and evaluate the barriers and potential consequences of autonomous driving.

The methodology has been proven effective in determining the key factors surrounding the introduction of autonomous vehicles.

Various studies conducted on the implementation of autonomous vehicles predict that fully autonomous vehicles will be available for the public in the 2020s, with their use on public roads becoming legal, but only few studies are included as references of this study.

Some recommendations (R) to improve the paper are as follows:

R1. Mention the relevant studies where some factors considered in this study has been defined and considered in evaluation of the autonomous vehicles.

R2. List of references must contain recent papers and studies published in the last three years in order to comparatively define and evaluate the actual barriers and potential consequences of autonomous driving.

R3. Include a Discussion section where all about may be included.

For example, see the references:

10.1007/s11116-017-9786-y

https://doi.org/10.3390/su11030588

10.1007%2F978-3-030-24067-7_35

10.1109/ACCESS.2018.2868339

https://doi.org/10.3390/su10093118

10.1109/ROBIO.2017.8324756

 https://doi.org/10.3390/socsci7030034

etc.

Author Response

The authors would like to thank the reviewer its valuable comments. All of them have been considered and the paper has changed accordingly. All changes and added text have been highlighted in red. Below you can find the responses to each comment

R1. The factors considered in this analysis, and used as a starting point were mentioned in Table 1. But this was not clear in the previous version of the paper. We have added some factor and clarify this (in Table 1, lines 149-151). We hope it is more clear in the new version of the paper.

R2. We have added new references (references number 6,8,9,10,24,25,26,28, in the list of references- lines 340-400). These references include the ones proposed by the reviewer and others about autonomous driving and focus groups. Four of these 8 references correspond to recent studies, published before 2017). The authors would like to thank the reviewer for the very interesting studies recommended.

R3. A new section has been added according to this comment. (Section 4: Discussion of Results, in lines 263-281). This section includes all the reflections mentioned before

Round 2

Reviewer 2 Report

the authors have responded to all comments